# Fabrication and Characterization of Gelatin/Polyvinyl Alcohol Composite Scaffold

**DOI:** 10.3390/polym14071400

**Published:** 2022-03-30

**Authors:** Mengwen Sun, Yajuan Wang, Lihui Yao, Ya Li, Yunxuan Weng, Dan Qiu

**Affiliations:** 1School of Material and Chemical Engineering, Ningbo University of Technology, Ningbo 315211, China; smwwen@163.com (M.S.); yaolh@nbut.edu.cn (L.Y.); liya@nbut.edu.cn (Y.L.); 2Beijing Key Laboratory of Quality Evaluation Technology for Hygiene and Safety of Plastics, Beijing Technology and Business University, Beijing 100048, China; wyxuan@th.btbu.edu.cn

**Keywords:** polyvinyl alcohol, gelatin, porous scaffold

## Abstract

In this study, porous scaffold materials based on polyvinyl alcohol (PVA) and gelatin (Gel) were successfully fabricated and characterized. The mechanism of the reaction, morphology, and crystallinity were investigated by Fourier transform infrared spectroscopy (FTIR), scanning electron microscopy (SEM), X-ray diffraction (XRD), and X-ray photoelectron spectroscopy (XPS). In addition, thermogravimetric analysis (TGA) was performed together with differential scanning calorimetry (DSC) for examining the thermostability and phase transformation of the scaffolds. Degradation and swelling studies of PVA/Gel composite scaffold materials were performed in phosphate-buffered saline. Finally, the mechanical performances had been determined. According to the results, the polymer matrix that was formed by the combination of PVA and gelatin had better thermal stability. The synthesized composite scaffold was amorphous in nature. The addition of gelatin did not affect the fishbone-like microstructure of PVA, which ensures the excellent mechanical properties of the PVA scaffold. The denaturation temperature and elastic modulus of the PVA scaffold were improved by the gelatin addition, but the physical and chemical properties of the PVA scaffold were weakened when the gelatin content exceeded 10%. In addition, the PVA-10G sample has suitable degradability. Therefore, the PVA/Gel composite scaffold might potentially be applied in the field of tissue engineering that demands high strength.

## 1. Introduction

Tissue engineering using biocompatible materials is a potential way to repair cartilage damage or bone tissue defects. In this application, three important components, namely, scaffold materials (namely, the 3D polymeric matrix), cells, and growth factors, combine together to affect the regeneration of the damaged organs or tissues [1,2,3]. Among these three components, scaffold materials are the most important factor that can provide an appropriate substrate for cells’ attachment, which can promote some anchorage-related events, such as matrix synthesis [4,5]. According to recent studies, tissue engineering materials with 3D porous structures not only provide the support for conglutination and germination of the cell but can also adjust the physiological activity of the cell effectively [6,7,8]. Therefore, polymeric materials with a porous structure have been utilized to replace traditional materials in scaffolds, which can be ascribed to their high compressive resistance, good biocompatibility, low cost, and low weight. A number of natural and synthetic materials, such as polylactic acid (PLA), polyglycolic acid (PGA), polycaprolactam (PCL), polyether glycol (PEG), chitosan, collagen, and alginate, have been fabricated to perform diversified structural, optical, and mechanical applications [9,10,11]. In this research, polyvinyl alcohol (PVA) was selected due to its excellent properties, including high optical transmission, water solubility, non-toxicity, biocompatibility, thermal stability, and non-corrosiveness, which make it a good matrix for mechanical and a variety of other applications [12,13]. According to the literature, PVA has already been utilized in scaffolds [14,15,16,17]. However, a scaffold fabricated from a single polymer would be inefficient to maintain all the required characteristics. Therefore, composite scaffolds created by tailoring two or more polymers have attracted increased attention in the present day.

Gelatin (Gel), a partial collagen hydrolysate comprising the Gly-Pro-Hyp sequences, can be discovered within the connective tissues under normal conditions. The gel, which is obtained from organisms, has several advantages, such as bioactivity, biocompatibility, biodegradability, and low antigenicity, and thus can be extensively applied in numerous fields, including cosmetics, medicines, and foods [18]. Because of their food safety, gelatin-based films play an important role in the preparation of sustained-release films, edible sausage casings, and capsule coatings [19,20,21]. In addition, gelatin is responsible for cell adhesion, proliferation, and differentiation. In addition, the abundance of active groups or reactive sites in gelatin ensures its high reactivity with other polymers [22,23]. Nevertheless, gelatin films have some disadvantages; for example, their low mechanical properties, rapid degradation rate, and low thermostability limit their applications.

Several biodegradable polyester scaffolds have been prepared and applied in biomedical fields. Though some studies reported that Gel and PVA can be prepared to fabricate scaffolds via the 3D printing technique [24], no one has reported information about the effects of the gelatin content on the properties of PVA-based composite scaffolds. However, this information might be useful for skin or soft tissue engineering. The present study mainly focused on the preparation of PVA-Gel composite scaffold to be applied in typical tissue engineering fields. The thermal stability and crystallinity were evaluated by means of DSC, TGA, and XRD. The pores and morphology were assessed using SEM. Phosphate-buffered saline (PBS) was used to study in vitro characteristics, such as swelling and degradation. Finally, this study expected to find a composite scaffold with relatively good performance by adjusting the amount of gelatin and attempted to evaluate whether the prepared composite scaffold can be used in skin tissue engineering.

## 2. Materials and Methods

### 2.1. Materials

Polyvinyl alcohol (PVA, with the polymerization and alcoholysis degrees of 1700 and 88%, respectively) was obtained from Aladdin Reagent Company (Shanghai, China). Gelatin (Type A) was purchased from Sigma-Aldrich. All other chemicals used in this study were analytically pure and obtained from Fisher Scientific Co. (Pittsburgh, PA, USA). 

### 2.2. Preparation of PVA/Gelatin Composite Materials

A precise amount of PVA (4.00 g) was slowly added to 40 mL of distilled water, and the solution was kept at 25 °C for 2 h to ensure complete dispersion and swelling; then, the solution was stirred at 90 °C for 30 min to obtain a homogeneous solution. A precise amount of gelatin (ranging from 0 wt.% to 30 wt.% based on the weight of PVA) was added to the same solution and stirred at 600 rpm for 4 h at 80 °C. Then, the system was subjected to centrifugation for 10 min at 3000 rpm to remove the trapped gas. Thereafter, the solution was slowly added to a square Petri dish and preserved in a refrigerator overnight. Afterward, each frozen sample was freeze-dried for three days at −50 °C. Finally, PVA-0G, PVA-5G, PVA-10G, PVA-20G, and PVA-30G (the number represents the content of gelatin) scaffolds were successfully prepared.

### 2.3. Characterization of PVA/Gelatin Composite Scaffold

Fourier transform infrared (FT-IR), X-ray diffraction (XRD), X-ray photoelectron spectroscopy (XPS), differential calorimetric scanning (DSC), and thermogravimetry (TG) were carried out to characterize the as-prepared composite scaffolds. An X-ray diffractometer (Bucker, D8 advance X-ray diffraction meter, Massachusetts, American) was used for determining the XRD patterns. The following parameters were set for sample detection: scanning speed, step size 0.01°, dwell time 0.1s per step; 2θ range, 5~60°; step size, 0.01°; tube current, 100 mA; working target voltage, 40 kV; radiation, Ni-filtered CuK α radiation; and wavelength λ, 1.5406 Å. Thermal stability analysis was recorded by DSC (German NETZSCH 204DSC, Netzsch instrument manufacturing Co., LTD, Bavaria, Germany) and TGA (Perkin-Elmer, Pyris I, Perkinelmer Inc., California, American) to investigate the thermal degradation of the composite scaffolds. The 3.0–5.0 mg samples were sealed within the DSC sample pan and heated at a rate of 10 °C/min until the temperature reached 300 °C (from 30 °C); the empty pan was used as the control. In TGA, the samples were scanned at a rate of 20 °C/min at 50–700 °C under a nitrogen atmosphere. In addition, Thermo Nicolet FT-IR (Nexus, Woodland, CA, USA) spectroscopy was utilized for the chemical study of each sample within the range of 4000–400 cm^–1^. An Axis Ultra Dld (Shimadzu, Kyoto, Japan) was used for XPS measurements with the implementation of an Al Ka X-ray source.

### 2.4. Mechanical Test

We further investigated the mechanical performances of the scaffold materials by assessing the elasticity modulus, and by carrying out an elongation test at the break value. The CMT4204 Tensile Machine (MTS Industrial Systems Co., LTD, Shanghai, China) was utilized to record the stress–strain curves for the films having strip shapes that were balanced in a dryer for 24 h at a constant temperature. In addition, a manual micrometer (resolution, 0.01 mm) was used for testing the thicknesses of the specimens. Thereafter, the tensile machine was used for griping every 8 × 2 cm^−2^ specimen (8 × 2 cm^−2^ for length × width) by its two jaws, and further, sample stretching was performed in the perpendicular direction at a constant rate of 20 mm/min. Finally, we determined the tensile strength together with the elongation at break values. All tests were carried out in triplicate to calculate the average value.

### 2.5. Scanning Electron Microscopy (SEM)

A scanning electron microscope (SEM, Hitachi S4800, Hitachi, Ltd., Tokyo, Japan) was used for investigating the morphology of the specimens at a working voltage of 10 kV. Each specimen was subjected to sputter coating with a 2.0 nm aurum; then, images were taken at an accelerated voltage of 10 kV.

### 2.6. In Vitro Swelling Characterizations

Phosphate-buffered saline (PBS, pH 7.4) was adopted to detect PVA/G scaffolds for their swelling features at 37 °C. Firstly, an appropriate amount of dried scaffold material was soaked within the PBS buffer; after that, the scaffold was drawn from the buffer, the non-absorbed buffer was eliminated using the filter paper, and the weight was measured every 2 min. Formula 1 was utilized to calculate the swelling ratio.
(1)$$SR\;=\;\frac{W_{t}\;-\;W_{0}}{W_{0}}\;\times \;100\%$$
where *SR* is the swelling ratio; *W_t_* is the wet weight of the sample; and *W*_0_ is the dry weight of the sample.

### 2.7. In Vitro Degradation of Composite Scaffold

To determine their degradation resistance, the PVA/Gel scaffolds were incubated using simulated body fluid. The specimens (*M*_0_) were weighed precisely and soaked in 10 mL of simulated body fluid; then, incubation was performed by placing the specimens in a digital biochemical incubator for several weeks at 37 °C. Next, the specimens were extracted at intervals of one week and rinsed with the simulated body fluid thrice. Afterward, each scaffold was placed in the oven for vacuum drying under ambient temperature until reaching constant weight (*M_d_*). Later, Formula 2 was applied for determining the ratio of specimen degradation.
(2)$$L_{s}(\%)\;=\;\frac{M_{0}\;-\;M_{d}}{M_{0}}\;\times \;100\%$$

### 2.8. Statistic Analysis

SAS (v.9.4, SAS Institute, Cary, NC, USA) was utilized for statistical analysis. The data obtained from the experiments were analyzed through analysis of variance (ANOVA).

## 3. Results

### 3.1. FTIR Analysis

FTIR spectra were used to characterize the structures of PVA/Gel scaffolds (Figure 1). In the PVA scaffold sample (without gelatin), a broad band appeared at 3425 cm^–1^ for the stretching vibration of O-H (ν-OH) having a strong intramolecular hydrogen bond, and the absorption band at 2950 cm^–1^ was obtained from the C-H stretching vibration (ν-CH) of the alkyl groups; the results are similar to those from Kim’s report [24]. Furthermore, the weak band at 1590 cm^–1^ represented the C=O stretching vibration (ν-C=O) due to the presence of a small quantity of polyvinyl acetate in the PVA molecule. For the protein, the infrared absorption characteristic peak of the amide I band (C=O stretching vibration, νC=O), the amide II band (N-H bending vibration, δN-H), and the amide III band (C-N stretching vibration, νC-N) appeared at around 1650, 1530, and 1230 cm^–1^ [25]. According to the literature, the O-H absorption band shifts toward the lower frequencies due to the formation of hydrogen bonds between and within the molecules [26]. As seen in Figure 1, it was noted that the band intensity at 1645 cm^–1^ increased for PVA/Gel composite scaffolds with the increase of the gelatin content, which revealed the formation of PVA-Gel intermolecular hydrogen bonds within those composites. Compared to the spectra of PVA and composite scaffolds, no new peak could be detected, suggesting that the composite scaffolds possibly contributed to forming the hydrogen bonds between and within the PVA-Gel molecules, thereby leading to the detected shift in the peaks. These results are consistent with the results of the XRD analysis.

### 3.2. XPS Analysis

Composite scaffolds containing different gelatin ratios were subjected to XPS for a better understanding of their chemical structures. The XPS spectra of the PVA and PVA-10G scaffolds are presented in Figure 2a. Peaks at C 1s and O 1s were observed in the spectra that belong to the PVA scaffold, while for PVA-10G, the scaffold peaks were detected at C 1s, O 1s, and N 1s. Figure 2b–g display the high-resolution curve-fitted XPS spectra for both samples. Three diverse carbon conditions were observed in C 1s spectra for the PVA scaffold (Figure 2b), which were hydrocarbons (C-C and C-H) at 284.6 eV, and a carbonyl (C=O) associated with the polyvinyl acetate at 288.4 eV. Nonetheless, for the PVA/Gel scaffold, a newly formed weak peak was detected at 286.3 eV (Figure 2e), which corresponds to the amine (C-N) bond of the gelatin part. As shown in Figure 2c, one peak at 533.2 eV was detected at the O 1s region of PVA, which belongs to the carbonyl group (C=O) [27]. After compounding with gelatin, O 1s showed an almost unchanged binding energy or peak, which can be confirmed by comparing Figure 2c with Figure 2f. These findings are consistent with the results of the FTIR.

Figure 3 presents the XPS spectra for the C 1s, O 1s, and N 1s peaks of the PVA/Gel composite scaffolds. As shown in Figure 3a, the peak of the amine bond (C-N) appeared at 286.5 eV. PVA-5G and PVA-10G had a stronger signal than PVA-20G and PVA-30G, indicating that: (1) it was not just a physical mixing between PVA and gelatin; (2) when the gelatin ratio exceeded 10%, it could not be combined with PVA. A similar tendency was found for the N 1s spectrum in Figure 3b; the peak intensity was weak for the PVA-20G sample. On the contrary, the N 1s signal was higher, and as a result, it can be presumed that the residual gelatin was not completely removed in PVA-30G.

### 3.3. XRD Analysis

The X-ray diffractions of all PVA-based scaffolds are represented in Figure 4. From the XRD patterns, a broad peak at 19.1° could be observed for the pure PVA scaffold. There were no obvious differences between the five scaffold materials. However, the broadness of the diffraction peak of the composite scaffolds increased while its intensity reduced as the concentration of gelatin increased, suggesting that the PVA/Gel scaffolds are amorphous. The above observations can be explained by the Hodge criterion that constructs the relationship between the peak height and the degree of crystallinity [28]. In addition, the air side of the spongy sheet hindered the molecular orientation as the moisturecuring occurred together with fast solvent evaporation in the process of freeze drying [29]. There was no new peak for the composites, which indicates that PVA was sufficiently compatible with gelatin; such results further prove the absence of a new phase within the as-synthesized polymeric matrix system.

### 3.4. SEM

The samples were freeze-dried before the SEM investigation, and the SEM images of the cross-section of all samples are shown in Figure 5. Similar to the reported literature, a fishbone-like structure could be found for the PVA scaffold [30,31]. In this work, the widths of the vertical direction were 1.5 to 5 um for the 10 wt.% PVA scaffold, which is consistent with the report of Zhou et al. [32]. The microstructure of PVA and gelatin composites showed similar morphologies; a slight difference was observed compared to the pure PVA scaffold when the gelatin content was lower than 20%. However, the regular fishbone-like structure disappeared for the PVA-30G scaffold material, and aggregation could be observed. Nevertheless, no obvious phase separation was observed for all gelatin-based PVA composite scaffolds, indicating the good compatibility between PVA and gelatin. The findings are consistent with the conclusions of XRD and FTIR. Compared to the five samples, PVA-10G displayed an indispensable and regular structure for tissue engineering materials.

### 3.5. Thermal Properties of Scaffold Materials

The DSC curves in Figure 6 show the phase transformation temperature of all PVA-based composites. According to this figure, two endothermic peaks were found for the PVA scaffold, probably because PVA is the alcoholysis product of poly(vinyl acetate) in this experiment. The endothermic peak at 115 °C was the phase point of PVAC [33], which is also consistent with the results of FT-IR. The phase transformation of the PVA-G composite scaffold had a similar tendency to that of pure PVA; however, the peak temperature and the enthalpy shifted to a higher value at a high temperature (about 195 °C, corresponding to the denaturation temperature of PVA) [34]. Therefore, it can be assumed that the addition of gelatin can enhance the thermal stability of PVA. During the composite process of the scaffolds, PVA-10G had a higher denaturation temperature.

The thermal degradation of each scaffold material was analyzed by TGA (Figure 7). The temperature corresponding to 50% weight loss (T_50_) is called the semi-life temperature of the polymer, and a higher T_50_ indicates better thermal stability of the polymer [35]. The T_50_ and ash data are described in Table 1. As shown in Table 1, the PVA-10G sample presented the highest T_50_ (391.72 °C), followed by the PVA-5G sample; both of these T_50_s were higher than that of PVA. These results indicate that the addition of gelatin could improve thermal stability. However, the thermal stability was decreased when the gelatin content exceeded 20%, which means that the stability of gelatin was lower than that of the polymer matrix formed from gelatin and PVA.

### 3.6. Swelling Properties

The ability of a scaffold to retain its liquid form is an important feature to evaluate its property for tissue engineering. The swelling properties of bio-macromolecules can be affected by their structure, their condition, the flexibility of their chain segment, and the interactions between the molecules [36]. The equilibrium swelling ratios of different PVA scaffolds are shown in Figure 8.

PVA is very sensitive to water due to the presence of many hydroxyl groups. On the other hand, gelatin is the hydrolysate of collagen that contains abundant hydrophilic groups, so it swells in an aqueous solution. According to Figure 8, the equilibrium swelling ratio of the composite scaffold materials was lower than that of PVA when the content of gelatin was less than 10%. However, the equilibrium swelling ratio might exceed 100% with the increase in the gelatin content. For this reason, it could be concluded that the proper amount of gelatin could form hydrogen bonds with the PVA molecules and reduce the number of free hydrophilic groups [36]; thus, the free volume for water molecules and reactive sites is also reduced, which leads to a lower swelling ratio. The large number of hydrophilic groups (e.g., amino and hydroxyl groups) that existed in the excess gelatin caused the increase in the equilibrium swelling ratio when the content of gelatin exceeded 10%. The higher water absorption capability of the scaffold material makes it more capable of absorbing the cell culture medium; thus, it can provide more nutrition for cell growth. On the other hand, if the material has ahigh water absorption property but obvious swelling, the scaffold may deform and the porous structure can be destroyed or even disappear, which is not conducive to cell growth. Nevertheless, the materials with moderate water absorption and a suitable swelling property would be ideal for tissue engineering applications [37]. The PVA-10G sample could meet the requirements of tissue scaffold materials.

### 3.7. Degradation Properties

Degradable tissue engineering scaffolds require a proper degradation cycle. On the one hand, the degradation cycle can facilitate cell expansion and tissue formation when the tissue is not sufficiently mechanically loaded. On the other hand, the scaffolds can be completely degraded to achieve the final repair and replacement after the formation of tissue. Therefore, degradation speeds that are too slow or too fast are not conducive to the repair and regeneration of tissues [38,39]. For the in vitro degradation, the weight reductions of the scaffolds were measured once a week for 15 weeks. Figure 9 shows the degradation properties of the PVA and PVA-G composite scaffold materials in simulated body fluid.

It could be seen that all PVA-based scaffold materials revealed excellent degradability. According to the results in Figure 9, all the samples were completely degraded within 7 to 10 weeks. Additionally, the weight loss ratio reached (95.4 ± 2.5) % after ten weeks. The macromolecular chains of PVA and gelatin were prone to be degraded by the enzymes in the presence of water. According to the literature, the new biological tissue forms within 70~105 days, which needs the scaffold materials to have a suitable degeneration speed [40]. Therefore, the composite scaffold based on PVA and gelatin shows potential as a degradable tissue engineering scaffold material.

### 3.8. Mechanical Properties

Table 2 lists the elasticity modulus and elongation at break of the dry materials. As suggested by the research of Fan et al., the interaction of the polymers might have a certain effect on the composite polymer concerning its mechanical performances [41]. The elasticity modulus increased with increasing gelatin content, but the elongation at break exhibited an opposite tendency. When the gelatin content was 5%, the elasticity modulus increased slightly from 91.88 MPa to 92.42 MPa and then sharply reached 247.55 MPa for PVA-10G. The significant increase in the elasticity modulus should originate from the cross-linking reaction between PVA and gelatin, as well as in the distinct and regular microstructure (see Figure 5). The attenuation of elongation at break should come from the fragility of gelatin.

## 4. Conclusions

In conclusion, poriferous 3D scaffolds based on polyvinyl alcohol (PVA) and gelatin were successfully fabricated. The effect of the gelatin content on the PVA/G composite scaffolds, in terms of thermal stability, morphology, swelling behavior, degradation, and crystallinity, was studied. The FTIR results reveal that intermolecular hydrogen bonds formed between PVA and gelatin in the composites. XRD and SEM showed the excellent miscibility of the two components. The content of gelatin had a prominent influence on the microstructure, thermal stability, equilibrium swelling behavior, and mechanical properties of the PVA/G blending scaffolds. At the beginning, the addition of gelatin improved the denaturation temperature and elasticity modulus of the PVA scaffold, but its physicochemical performance became weakened when the gelatin content exceeded 10%. Hence, the results of the thermal stability, swelling properties, and mechanical performance reveal that PVA-10G might be suitable for tissue engineering applications. Therefore, we will continue this research and hope to obtain excellent biomedical materials in terms of biocompatibility, cell adhesion, and hemolysis through in vivo research.

## Figures and Tables

**Figure 1 polymers-14-01400-f001:**
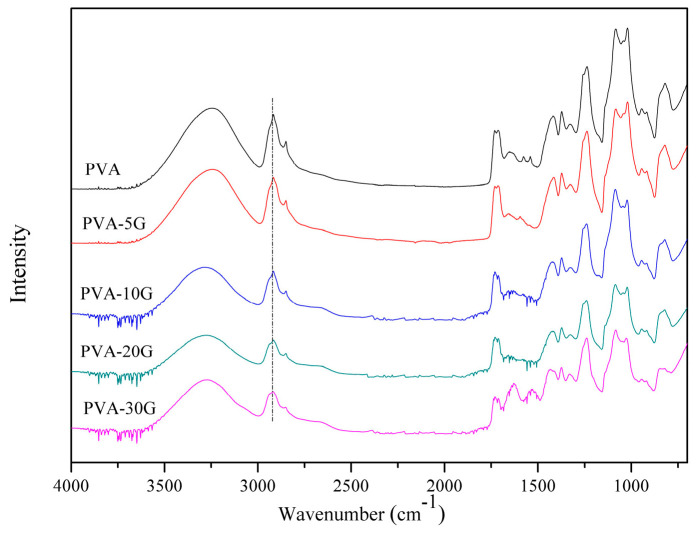
FT-IR spectra of all samples.

**Figure 2 polymers-14-01400-f002:**
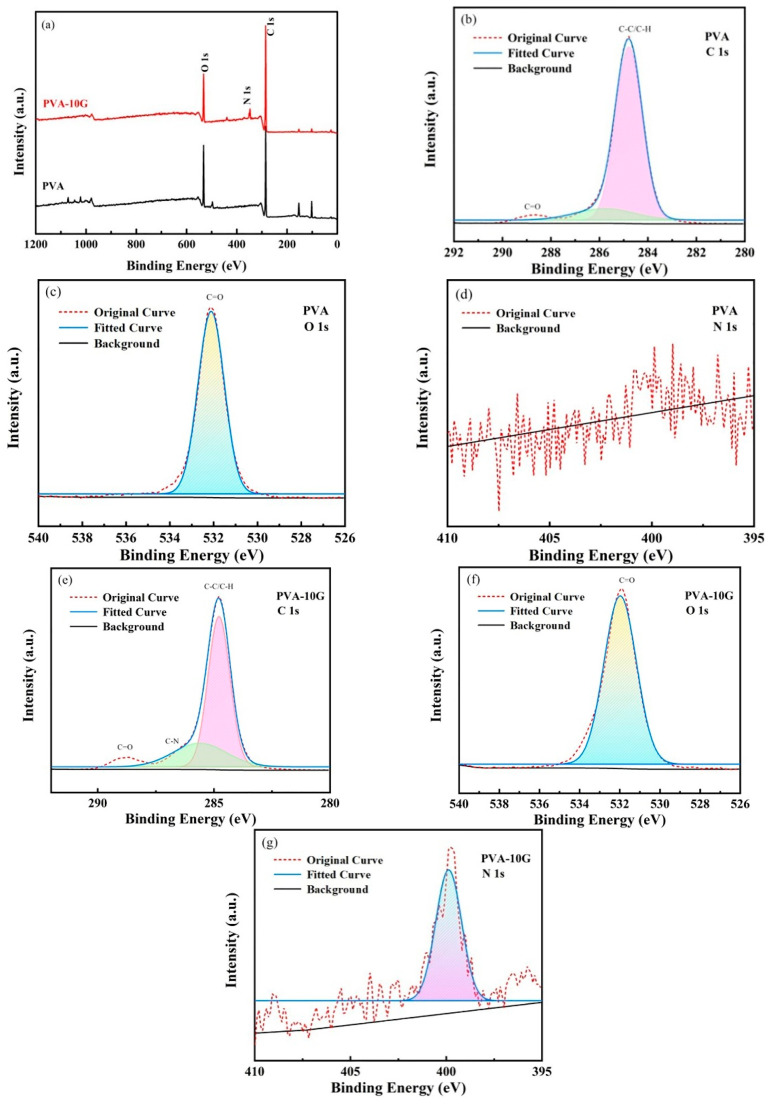
The measured XPS spectra of the PVA and PVA-10G scaffolds. (**a**) XPS spectra of PVA and PVA-10G scaffolds; (**b**) PVA C 1s; (**c**) PVA O 1s; (**d**) PVA N 1s; (**e**) PVA-10g C 1s; (**f**) PVA-10g O 1s; (**g**) PVA-10G N 1s.

**Figure 3 polymers-14-01400-f003:**
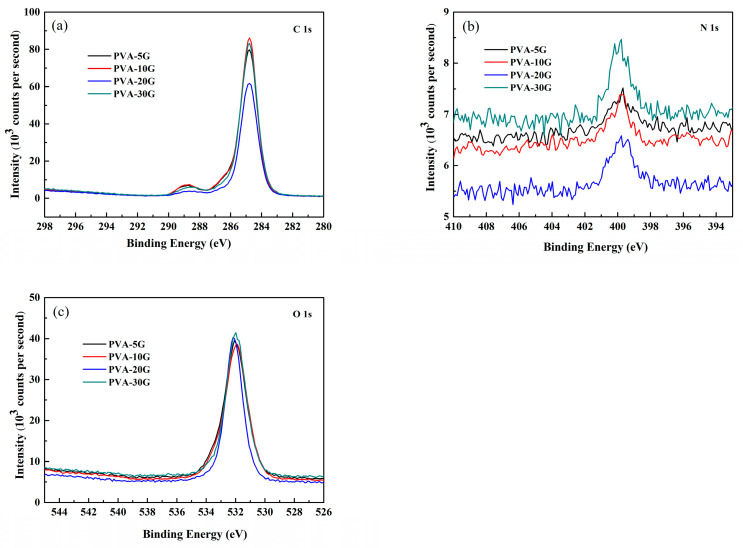
The XPS spectra of the PVA/Gel composite scaffolds:(**a**) C 1s, (**b**) N 1s, and (**c**) O 1s.

**Figure 4 polymers-14-01400-f004:**
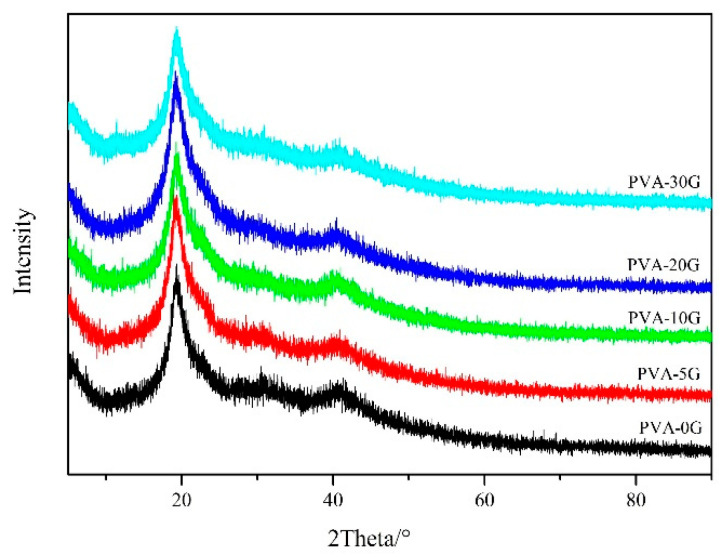
XRD patterns of PVA-based scaffolds.

**Figure 5 polymers-14-01400-f005:**
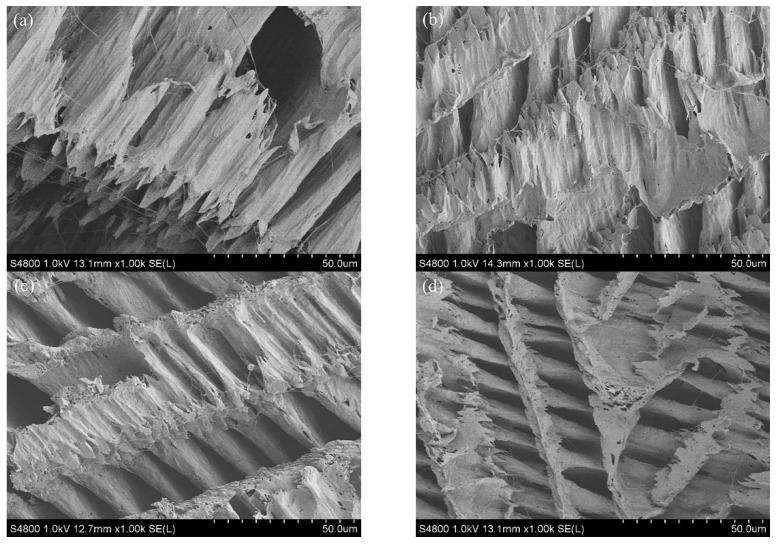
SEM images showing the scaffold inner structure after freeze drying in PVA solutions with diverse gelatin contents: (**a**) PVA; (**b**) PVA-5G; (**c**) PVA-10G; (**d**) PVA-20G; (**e**) PVA-30G.

**Figure 6 polymers-14-01400-f006:**
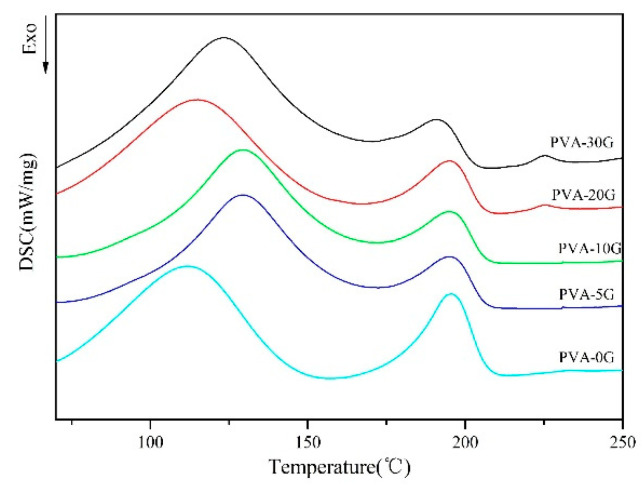
DSC thermograms of PVA/G blend scaffolds at a heating rate of 10 °C/min.

**Figure 7 polymers-14-01400-f007:**
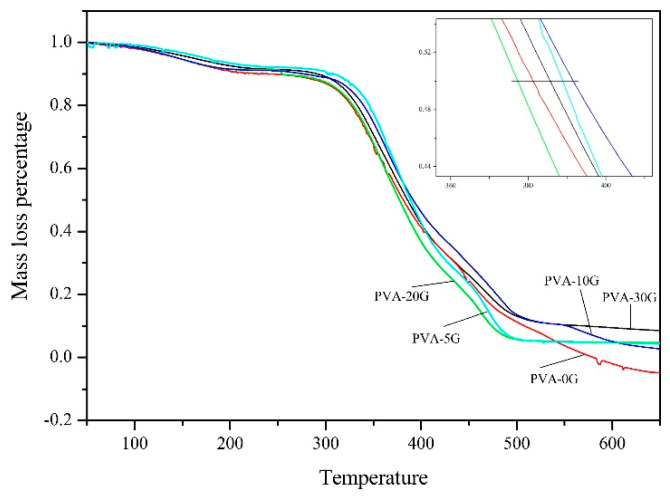
TGA patterns of PVA and PVA/Gel scaffolds.

**Figure 8 polymers-14-01400-f008:**
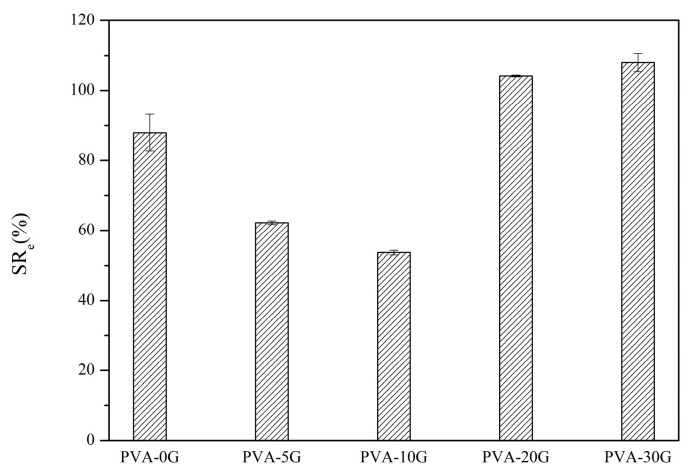
Equilibrium swelling ratio of PVA/G composites.

**Figure 9 polymers-14-01400-f009:**
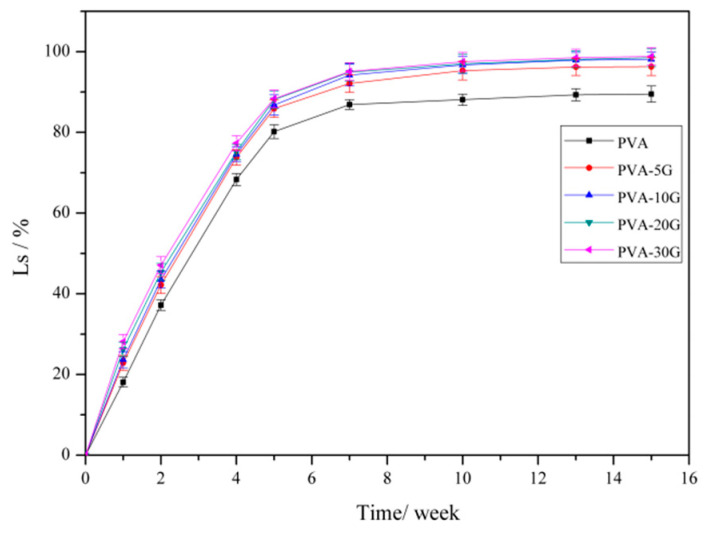
Change in the weight of samples at different degradation times.

**Table 1 polymers-14-01400-t001:** Temperature for T50 and ash yield of PVA and PVA/G composite scaffolds.

Sample	T_50_/°C	Remaining Yield/%
PVA-0G	381.57	0.01
PVA-5G	388.86	4.33
PVA-10G	391.72	2.67
PVA-20G	377.64	4.59
PVA-30G	385.92	8.53

**Table 2 polymers-14-01400-t002:** Mechanical properties of scaffold materials.

Sample	Elasticity Modulus/MPa	Elongation at Break/%
PVA-0G	91.88 ± 14.04	170.73 ± 13.42
PVA-5G	92.42 ± 13.08	186.19 ± 15.79
PVA-10G	247.55 ± 33.12	126.95 ± 17.92
PVA-20G	241.77 ± 17.17	59.30 ± 3.96
PVA-30G	215.26 ± 30.19	47.44 ± 7.54

## Data Availability

The data that support the findings of this study are available within the article.

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
