# Peer review of "Fabrication and Characterization of Gelatin/Polyvinyl Alcohol Composite Scaffold"

_polymers, 2022, doi:10.3390/polym14071400_

Round 1

Reviewer 1 Report

The authors have made an interesting article describing the fabrication and characterization of Gelatin/Polyvinyl Alcohol composite scaffold for potential tissue engineering applications but some improvements should be made in order to be published in Polymers:

  1. In the introduction the term “superb” should be replaced to a more suitable one, from the expression: “ascribed to the superb physiochemical performances, cost-effectiveness, and low weight” (line 38-39).
  2. At line 62-64 please rephrase the whole phrase because it is not accurate: “Though some studies reported the preparation of scaffolds from Gel and PVA via the 3D printing technique,24 no one reported the information about the effects of gelatin content on the properties of solution-blended PVA-based composite scaffold”
  3. Figure 1 representing the FTIR spectra should be replaced or redone because it lacks clarity.
  4. When referring to the FTIR analysis from results please write “cm-1” each time and not “cm-1” -1 should be superscript.
  5. Also the Figure 6, Figure 7, Figure 8 and Figure 9 lacks clarity please replace them with more suitable ones.

Author Response

Dear Reviewers:

     We wish to express our appreciation to the Reviewers for their insightful comments, which have helped us significantly to improve our manuscript. According to the suggestions, we have thoroughly revised our manuscript and its final version is enclosed. Point-by-point responses to the comments are listed below (in blue).

  1. In the introduction the term “superb” should be replaced to a more suitable one, from the expression: “ascribed to the superb physiochemical performances, cost-effectiveness, and low weight” (line 38-39).

## Response: Thank you for your constructive suggestions. We've replaced the corresponding words with relatively technical descriptions. We specifically describe excellent performance as strong resistance to stress and good biocompatibility. (line 39-41)

  1. At line 62-64 please rephrase the whole phrase because it is not accurate: “Though some studies reported the preparation of scaffolds from Gel and PVA via the 3D printing technique,24 no one reported the information about the effects of gelatin content on the properties of solution-blended PVA-based composite scaffold”

## Response: Thank you very much for your comments. We have rewritten the sentence as ‘Though some studies reported that Gel and PVA can be prepared to fabricate scaf-folds by 3D printing technique, no one reported the information about the effects of gelatin content on the properties of PVA-based composite scaffold’. (line 70-72)

  1. Figure 1 representing the FTIR spectra should be replaced or redone because it lacks clarity.

## Response: Thank you very much for pointing out this problem. We have re-uploaded the new figure. (Figure 1)

  1. When referring to the FTIR analysis from results please write “cm-1” each time and not “cm-1” -1 should be superscript.

## Response: Thank you very much for pointing out the error. We are very sorry for this and have rewritten the unit symbol in the paragraph. (line 168-179)

  1. Also the Figure 6, Figure 7, Figure 8 and Figure 9 lacks clarity please replace them with more suitable ones.

## Response: Thank you very much for pointing out this problem again. We are very sorry for this and have re-uploaded all images. (Figures)

Reviewer 2 Report

Although the topic is interesting, some comments appended below should be addressed prior to accepting the publication of this manuscript as follow:

  1. The abstract should be improved by highlighting the significant findings.
  2. Some biological tests should be conducted to recommend the current candidate for introduction into biomedical applications, such as hemolysis and cellular toxicity tests.
  3. The entire figures should be improved.
  4. In the introduction, please explain the novelty of this study at the end of the introduction.
  5. Some relevant articles could be used to enhance the quality of your discussion, such as https://doi.org/10.1038/s41598-021-82963-1).  

Author Response

Dear Reviewers:

     We wish to express our appreciation to the Reviewers for their insightful comments, which have helped us significantly to improve our manuscript. According to the suggestions, we have thoroughly revised our manuscript and its final version is enclosed. Point-by-point responses to the comments are listed below. (in yellow)

  1. The abstract should be improved by highlighting the significant findings.

## Response: Thank you for your constructive suggestion. We added the scaffold with a gelatine content of 10%, which was a turning point. When the gelatin content exceeds this ratio, the physical and chemical properties of the scaffold will be affected. (line 23-26)

  1. Some biological tests should be conducted to recommend the current candidate for introduction into biomedical applications, such as hemolysis and cellular toxicity tests.

## Response: Thank you very much for your comments. This study is ongoing, and we will evaluate its biomedical applications including in vitro cell experiments, hemolysis and toxicity tests in future studies. 

  1. The entire figures should be improved.

## Response: Thank you very much for pointing out this problem. We have re-uploaded the new figures.

  1. In the introduction, please explain the novelty of this study at the end of the introduction.

## Response: Thank you for your constructive suggestions. We included the purpose and expectations of the study in the introduction. We hope to find the composite scaffold with good comprehensive performance by the amount of gelatin, and also try to evaluate whether it has application in skin tissue engineering. (line 78-80)

  1. Some relevant articles could be used to enhance the quality of your discussion, such as https://doi.org/10.1038/s41598-021-82963-1).

## Response: Thank you for your comments. In this experiment, we obtained tissue materials with good mechanical properties and degradation properties, but their application in vivo still needs to be evaluated, so we added our follow-up research direction into the conclusion. (line 372-373)

Round 2

Reviewer 1 Report

I consider that the authors made the proper modification and the article can be published.  

Reviewer 2 Report

The authors have addressed all the comments carefully.

This manuscript is a resubmission of an earlier submission. The following is a list of the peer review reports and author responses from that submission.

Round 1

Reviewer 1 Report

In this manuscript, authors present the fabrication and characterization of porous scaffold materials based on polyvinyl alcohol (PVA) and gelatin (Gel).

The paper can be published in this form if the authors will made some corrections:

  • At Correspondence, modify “Email: qiudan_zju@163.com;Prof. WENG Yunxuan, Tel.:+86-10-68985535” with “Email: qiudan_zju@163.com; Prof. WENG Yunxuan, Tel.: +86-10-68985535”
  • At 2.3 section, modify “to 300℃ at a heating rate of 10℃/min” with “to 300 ℃ at a heating rate of 10 ℃/min
  • At 2.6 and 2.7 sections, modify “at 37℃” with “at 37 ℃”
  • At 2.7 section, indicate in formula 2 what is Ls
  • At 3.1 section, modify “1230 cm-1[22]” with “1230 cm-1 [22]”
  • In all manuscript, modify “FTIR”, with “FT-IR”
  • At 3.5. section, modify “peak at 115℃” with “peak at 115 ℃”
  • At 3.6 section, modify “applications[34]” with “applications [34]”
  • At Author Contributions and Funding, respectively, please delete “Author Contributions:” and “Funding:” because are duplicate
  • Write the references by the same format.

Reviewer 2 Report

Polymer, MDPI

Type: Article

Title: Fabrication and Characterization of Gelatin/Polyvinyl Alcohol Composite Scaffold

Authors: Yajuan WANG, Caikun XU, Lihui YAO, Ya LI, Dan QIU and Yunxuan WENG

Comments:

The topic and overall content of this research manuscript (MS) seem no exciting and no appealing to the scientific world. In particularly, this paper lacks novelty in terms of the materials and characterization approaches. It is nothing new from the use of gelatin/PVA scaffold even though the authors characterize it for several experiments.

For the novelty part of this paper: the background information does not give a vivid detail as to why this research was considered as a novel approach compared to others. Furthermore, their claim material’s fabrication and characterization do not fit the scientific and ethical publication – quite a high level of plagiarisms from previously published work, and no referencing is made (see the attached MS). The authors should have appropriately cited similar work done on the related topic. Therefore, I am sorry that the manuscript is recommend rejecting without any consideration.
